# Cation Vacancies in Feroxyhyte Nanosheets toward Fast Kinetics in Lithium–Sulfur Batteries

**DOI:** 10.3390/nano13050909

**Published:** 2023-02-28

**Authors:** Aimin Niu, Jinglin Mu, Jin Zhou, Xiaonan Tang, Shuping Zhuo

**Affiliations:** School of Chemistry and Chemical Engineering, Shandong University of Technology, Zibo 255049, China

**Keywords:** lithium–sulfur batteries, Fe vacancies, catalysis, kinetics

## Abstract

Lithium–sulfur batteries have attracted extensive attention owing to their environmental friendliness, abundant reserves, high specific discharge capacity, and energy density. The shuttling effect and sluggish redox reactions confine the practical application of Li–S batteries. Exploring the new catalyst activation principle plays a key role in restraining polysulfide shuttling and improving conversion kinetics. In this respect, vacancy defects have been demonstrated to enhance the polysulfide adsorption and catalytic ability. However, inducing active defects has been mostly created by anion vacancies. In this work, an advanced polysulfide immobilizer and catalytic accelerator is developed by proposing FeOOH nanosheets with rich Fe vacancies (FeVs). The work provides a new strategy for the rational design and facile fabrication of cation vacancies to improve the performance of Li–S batteries.

## 1. Introduction

With the wide application of fossil energy, environmental pollution and the energy crisis have become two major problems in the world [1]. In this context, it is of great significance to develop new energy devices based on electrochemistry. Electrochemical energy storage equipment that is clean and has a low cost, high efficiency, low pollution, and the ability to be recycled has attracted wide attention. The current energy storage technology is mainly chemical energy storage. Lithium-ion battery has been the leader in portable electronic devices, energy storage devices, and other applications thanks to its technical safety, high energy conversion efficiency, and economic advantages [2]. However, the energy density of conventional lithium-ion batteries is approaching its limit. Lithium–sulfur (Li–S) battery is one of the most promising secondary lithium batteries thanks to its superior theoretical specific capacity (1675 mAh g^−1^) and energy density (2600 Wh kg^−1^) [3]. Its low cost and environment friendly characteristics, as well as the natural abundance of sulfur, also make Li–S batteries more favorable for commercial applications [4]. However, Li–S batteries confront many challenges, including rapid decline in the capacity and inferior cycle life originated from the shuttle effect of soluble lithium polysulfides (LiPSs) and depressed redox kinetics [5].

To address these problems, researchers have focused on accelerating the conversion of LiPSs by catalysis. Polar materials have been extensively researched to adsorb and accelerate the conversion of LiPSs, such as metal sulfides, nitrides, oxides, and carbides. Yang et al. prepared an Fe-PGM hybrid (Fe_2_O_3_ hybrid) with uniform distribution of α-Fe_2_O_3_ nanoparticles on three-dimensional layered porous graphene, which serves as a novel sulfur host and achieves a high rate and cycle stability for lithium–sulfur battery cathodes [6]. It is found that α-Fe_2_O_3_ NPs not only has a strong interaction with polysulfide, but also promotes the conversion of soluble polysulfide into insoluble products during the charging and discharging process, which effectively inhibits the shuttle of polysulfide and improves the utilization rate of sulfur. Moreover, Sun et al. reported that composite materials containing MnS (MnS/CNFs) were synthesized by electrospinning technology as the interlayer of Li–S battery to improve the electrochemical performance [7]. MnS containing sulfophilic active sites can enhance the chemisorption of polysulfides, thus inhibiting the shuttle effect and accelerating the transformation kinetics of polysulfides. The experimental results showed that the battery with an interlayer can reduce self-discharge and improve the electrochemical performance. These materials exhibit stronger polar–polar chemical adsorption with LiPSs, which can inhibit the shuttle of polysulfides and improve the reaction kinetics [8]. As is known to all, the intrinsic catalytic activity of catalysts is related to the surface atomic and electronic structures [9,10]. To date, exposing more active sites has been developed to enhance the catalytic activity of the polar materials [11]. Defect engineering has gained substantial popularity owing to abundant edge defects and the exposure of active sites [12]. Scientists have found that cation and anion vacancies engineering is a convenient way to adjust the local electron structure to accelerate the conversion of polysulfide. Anion vacancy sites including oxygen vacancies (e.g., WO_3−x_ [13] and TiO_2−x_ [10,14]), sulfur vacancies (e.g., TiS_2−x_ [15] and MoS_2−x_ [16]), selenide vacancies (e.g., Sb_2_Se_3−x_ [17] and WSe_2−x_ [18]), phosphorus vacancies (e.g., BP_4−x_ [19]), and nitrogen vacancies (e.g., g-C_3_N_4−x_ [20]) present excellent catalytic performance to promote the conversion of polysulfide in redox reactions. Mai et al. modified the membrane by introducing oxygen vacancy on the surface of TiO_2_ and improved the high energy density of the battery [21]. By experiments and DFT calculations, the oxygen vacancy means titanium dioxide has strong adsorption of polysulfide and high conductivity. In addition, oxygen vacancy greatly improves the catalytic conversion ability of polysulfide. Wang et al. reported that sulfur vacancies can enhance the adsorption of polysulfides and thus accelerate the catalytic conversion kinetics [22]. Compared with anion vacancies, reports of cation vacancy sites are still lacking because of the obstacles of their higher formation energy [23]. 

In this study, considering that Fe is more plentiful and cheaper than other transition metal (e.g., Co, Mo, and Ni), we chose FeOOH nanosheets with abundant Fe vacancies (v-FeOOH) and in situ incorporated them into graphene and carbon nanotube substrates (rGO /CNTs) using a simple wet-chemistry method. As a comparison sample, FeOOH nanosheets without Fe vacancies named FeOOH were used. The carbon network works as a conductive skeleton. The nanosheet structure of FeOOH shorten the pathways for fast lithium ion diffusion and electron transmission. Fe vacancies (FeVs) in FeOOH nanosheets are negatively charged and can serve as Lewis base sites in the cathode. The Lewis base sites serve as a lithium ion anchor and soluble polysulfide accelerator through the strong S_x_Li...FeVs interactions, leading to good electrode kinetics. As a consequence, the v-FeOOH/rGO/CNTs @ S composite as an Li–S battery cathode shows a high initial specific capacity of 1645.5 mAh g^−1^ and a low capacity fading rate of 0.125% per cycle at 1 C over 800 cycles.

## 2. Materials and Methods

### 2.1. Synthesis of rGO/CNT

Firstly, graphene oxide (GO) was fabricated from natural graphite powder using a modified Hummers method [24]. CNT was purchased by Sinopharm Chemical Reagent Co., Ltd. Afterwards, the acidified carbon nanotubes and GO were added into the above solution according to the mass ratio of 1:1, and the probe was ultrasonic detected for 30 min. In order to obtain rGO/CNT, the GO/CNT by freeze drying was placed in the tube furnace and heated up to 800 °C at a rate of 5 °C min^−1^ under the protection of Ar for another 2 h.

### 2.2. Preparation of v-FeOOH /rGO/CNT@S Cathode

In a typical procedure, CTAB (5 mmol) was added to 50 mL of rGO/CNT solution in a 100 mL three-necked flask followed by purging with N_2_ for about 15 min. FeSO_4_·7H_2_O (1.5 mmol) was added into the flask. Afterward, a freshly prepared NaBH_4_ solution (1.5 mmol, 2 mL) was added drop-by-drop to the above solution. With the flow of N_2_, as a strong reducing agent, sodium borohydride reduced Fe^2+^ to produce a black Fe-B composite comprising iron crystallites embedded in an amorphous boron nanosphere matrix, which gradually grew on the rGO/CNT. After 40 min, N_2_ was turned off and the sample was placed in air for 4 h. After being washed with deionized water and ethanol three times, the v-FeOOH /rGO/CNT was obtained. The v-FeOOH/rGO/CNTs@S cathode materials were obtained through a typical melt-diffusion method. In detail, first, the mass ratio of S to host material was 60:40 and it was ground for 20 min until evenly mixed. The mixture was then transferred to a tubular furnace where it was heated in argon protection at 155 °C for 12 h.

### 2.3. Preparation of FeOOH/rGO/CNT@S Cathode

As a control sample, FeOOH without Fe vacancies (named FeOOH) based on rGO/CNT was obtained by the fast oxidation of Fe(OH)_2_ particles with H_2_O_2_. Specifically, 1.5 mmol of FeSO_4_·7H_2_O was added in 40 mL of rGO/CNT solution, and 10 mL of NaOH (10 M) was then transferred to the iron salt solution at 45 °C in N_2_ atmosphere. The resulting Fe(OH)_2_ film on rGO/CNT was reacted at 45 °C for 6 h, 5 mL H_2_O_2_ (30%) solution was instantly poured into the suspension, and then 10 mL H_2_O_2_ solution was later added. The products of the reaction were taken out and washed with deionized water and ethanol several times and freeze dried. The preparation of the FeOOH/rGO/CNT@S cathode is the same as above.

### 2.4. Density Functional Theory Calculations

The density functional theory (DFT) calculations were carried out with Dmol^3^ implemented in the Materials Studio 2019, using Perdew–Burke–Ernzerhof (PBE) generalized gradient approximation, double numeric polarized (DNP) basis set, and D2 dispersion correction. The FeOOH (001) surface was chosen for the calculation. The Fe-deficient FeOOH (001) surface was built by removing Fe, O, and H atoms from an intact FeOOH surface. A 2 × 2 supercell was used and the vacuum height was 15 Å. Convergence tests of Monkhorst–Pack grids were performed to ensure the results. The adsorption energies (Eads) of Li_2_S_6_ on the FeOOH (001) and Fe-deficient FeOOH (001) surface were calculated.

## 3. Results

Figure 1a schematically illustrates the synthetic process of v-FeOOH/rGO/CNTs @S composite. The synthesis details can be obtained in the Experimental Section (Appendix A). Figure 1b displays the scanning electron microscopy (SEM) image of v-FeOOH. Appendix A shows SEM of v-FeOOH/rGO/CNT, which indicates that the v-FeOOH samples were obtained in the form of nanosheets and grown uniformly on the surface of the carbon matrix. The energy dispersive spectroscopy (EDS) mapping of v-FeOOH /rGO/CNTs (Appendix A) clearly demonstrates the elemental uniformity of Fe, O, and C elements throughout the nanosheet. The result reveals that the FeVs are evenly decorated on the FeOOH nanosheets. In Appendix A, the transmission electron microscopy (TEM) image confirms that the v-FeOOH consists of a nanosheet architecture. In addition, the high-resolution TEM (HRTEM) images of v-FeOOH show obvious lattice distortions and a clear lattice spacing of 0.25 nm, which can correspond to the (100) face of FeOOH (Figure 1c–e) [25]. Figure 1f shows the TEM elemental mapping, demonstrating the even distribution of C, O, and Fe in the prepared composite materials. In contrast, Figure 1g,h shows the SEM and HRTEM images of FeOOH, which exhibited excellent crystallinity with the ordered lattice. The FeOOH samples were also grown uniformly on the carbon matrix, guaranteeing the conductivity of electrodes in Appendix A. The electron paramagnetic resonance (EPR) can further confirm the defective feature of v-FeOOH and FeOOH (Figure 1i). Prominent EPR signals can be tested at ~3000 G, attributed to the magnetic interactions of Fe^3+^ [26]. Meanwhile, the stronger EPR signal of v-FeOOH compared with that of FeOOH reveals the valence-state change of v-FeOOH. The X-ray diffraction (XRD) patterns prove that the crystal plane structure for FeOOH is assigned to JCPDS No. 81-0462 (Figure 1j) [27]. The lower crystallinity of v-FeOOH is proved by the broader and weaker XRD spectra than that obtained by FeOOH, which is also consistent with the HRTEM of the two composite materials.

In order to investigate the potential mechanism of the electrochemical improvements for the v-FeOOH catalyst, DFT calculations were performed. Appendix A show the optimized v-FeOOH and FeOOH structure. Figure 2a–c displays the optimized geometries of Li_2_S_6_ adsorption on v-FeOOH (001) and FeOOH (001) surfaces, and the neighboring location to Fe vacancies’ surfaces. Based on the calculations, Li_2_S_6_ presents a significantly higher adsorption energy of −4.72 eV on v-FeOOH than that on the surface of FeOOH (−4.05 eV) and the neighboring location to Fe vacancies’ surfaces, indicating the higher LiPSs adsorption capacity of v-FeOOH. This is the reason for the stronger chemical sulfur immobilization of v-FeOOH and the generated high-efficiency shuttle suppression. The results are also in accordance with the digital photos of LiPSs solution adsorption by v-FeOOH and FeOOH after 4 h and the corresponding UV/Vis spectra in Figure 2d. In the v-FeOOH UV/Vis spectra, the absorption intensity of Li_2_S_6_ solution containing v-FeOOH is the weaker within the visible range, which indicates that the content of Li_2_S_6_ in the solution is the lowest. Moreover, the color of Li_2_S_6_ solution is shown to be significantly clear in the v-FeOOH-containing solution. The results signify the strong LiPSs adsorbability of v-FeOOH. X-ray photoemission spectroscopy (XPS) was used to verify the surface compositions and chemical states of v-FeOOH and FeOOH (Appendix A and Figure 2e–h). As shown, the peak area ratio of Fe^3+^ to Fe^2+^ (I_3_/I_2_) of v-FeOOH is higher than that of FeOOH, suggesting the increased FeVs in v-FeOOH [28].

Figure 3a shows the cyclic voltammetry (CV) curves of the batteries with a voltage range of 1.7–2.8 V at a scan rate of 0.2 mV s^−1^. The peak potentials of v-FeOOH and FeOOH are shown in Appendix A. The CV results exhibit two typical distinct reduction peaks and one reversible oxidation peak. As shown in Figure 3a, it is obvious that the polarization voltage between the second cathodic and anodic peaks is 39 mV for the FeOOH electrode and 30 mV for the v-FeOOH electrode. This is well consistent with the galvanostatic charge–discharge curves, suggesting its lower electrochemical polarization and improved redox kinetics by the v-FeOOH electrode [29]. To understand the electrocatalytic activity of v-FeOOH, lithium ion diffusion kinetics analyses were performed. First, we performed CV curves with different scan rates ranging from 0.2 to 1.5 mV s^−1^ to verify the Li-ion diffusion coefficient ($D_{Li}^{+}$) of different electrodes (Figure 3b,c). The Li-ion diffusion coefficient can be calculated by the classical Randles–Sevick equation: $I_{P}=2.69\times 10^{5}n^{1.5}aD_{Li^{+}}^{0.5}v^{0.5}C_{Li}$. Herein, $I_{P}$ stands for the current peak, n represents the number of charge transfer, and a is the active surface area. $D_{Li}^{+}$ is the Li-ion diffusion coefficient, v is the scan rate, and $C_{Li}$ is the Li-ion concentration in the electrolyte [30]. Accordingly, the linear relationship of $I_{P}$/$v^{0.5}$ can evaluate the Li-ion diffusion capability as n, a, and $C_{Li}$ are constants in our battery system. It can be concluded that, the larger the fitting curve slope, the faster the diffusion of lithium ions, further demonstrating that the v-FeOOH electrode can effectively enhance the conversion of polysulfides. Here, as shown in Figure 3d, the slope (S) of the v-FeOOH electrode ($S_{I_{1}}$ = 5.29, $S_{I_{2}}$ = 1.92, $S_{I_{3}}$ = 2.72) is higher than that of FeOOH ($S_{I_{1}}$ = 2.07, $S_{I_{2}}$ = 0.71, $S_{I_{3}}$ = 0.38). The results indicate the fast ion diffusion and accelerated reaction kinetics on the v-FeOOH electrode.

To investigate the superiority of v-FeOOH as a sulfur accelerator for Li–S batteries, electrochemical evaluations were carried out. Figure 3e shows the galvanostatic charge–discharge curves with two typical discharge plateaus. The first plateau is around at 2.3 V, assigned to the conversion of sulfur to soluble long-chain LiPSs. Another plateau is at around 2.1 V, which agreed with the long-chain LiPSs to Li_2_S_2_ and/or Li_2_S. In comparison with FeOOH without the FeVs electrode, FeOOH with the FeVs electrode manifested a higher discharge capacity at the two discharge plateaus. It is noted that FeOOH with the FeVs electrode displays smaller polarization (ΔE = 273 mV) than other electrodes (ΔE _FeOOH particle/rGO/CNTs_ = 290 mV). The results not only verified the superiority of FeOOH with the FeVs electrode in restraining the shuttle effect, but also indicated that v-FeOOH contributed to promoting the catalytic conversion of polysulfide [31]. As shown in the discharge process (Appendix A), v-FeOOH exhibits a lower nucleation overpotential of 2.104 V in comparison with FeOOH (2.083 V). In the charge process, the curve of the v-FeOOH electrode displayed the smallest initial charge potential barrier, which indicated an excellent catalytic effect of the v-FeOOH electrode in the decomposition of Li_2_S (Appendix A) [32]. Figure 3f displays the rate performances of v-FeOOH and FeOOH electrodes with a current density range between 0.1 and 5.0 C and returning to 0.1 C (1.0 C = 1675 mA g^−1^). The v-FeOOH electrode shows reversible discharge capacities of 1642.6, 1278.7, 831.1, 637, 460.7, and 332.2 mAh g^−1^ at 0.1 C to 0.2 C, 0.5 C, 1 C, 2.0 C, and 5.0 C, respectively. As a contrast, the FeOOH electrode shows lower discharge capacities at the identical current densities, respectively. The charge–discharge profiles and polarization characteristic of the two samples in Appendix A showed reversible electrochemical processes at different current densities, which also indicated the fast redox kinetics and low polarization of the v-FeOOH electrode. The cycling performances of the v-FeOOH and FeOOH electrodes in Figure 3g show that the v-FeOOH electrode shows a high initial specific capacity of 1645.5 mAh g^−1^ and still has a specific capacity of 320.8 mAh g^−1^ after prolonged cycling for 800 cycles under a high current density of 1 C. The first cycle columbic efficiencies of the v-FeOOH and FeOOH electrodes are 97.95% and 92.06%, respectively (Appendix A). The results are much higher than that of the FeOOH electrode (947.1 mA h g^−1^ and 102 mA h g^−1^ for 500 cycles, respectively). Even when the current density increased to 5 C, the v-FeOOH electrode exhibits excellent cycling stability (Appendix A). After 300 cycles, it still retained a high specific capacity of 171 mAh g^−1^, accompanied by an average capacity fading rate of only 0.033% per cycle.

Symmetrical cells were further tested to explore the reaction behavior of polysulfides on the surface of v-FeOOH in Figure 4a. Compared with FeOOH (Appendix A), it can be clearly found that v-FeOOH features the highest current density at various scan rates. Even at 50 mV s^−1^, the v-FeOOH electrode exhibits the same trend. In Figure 4b, the two pairs of redox peaks in the CV curve attribute to the mutual conversion between Li_2_S_6_, Li_2_S_4_, and Li_2_S [33]. The good performance of the symmetrical cell of v-FeOOH is consistent with the low interface contact resistance of the symmetrical cell revealed by the EIS test (Appendix A). The result further confirms the rapid polysulfide redox conversion on the v-FeOOH electrode, which is attributed to its stronger LiPSs adsorption [34]. Tafel curves of the v-FeOOH and FeOOH electrodes, resulted from potentiostatic polarization experiments, are exhibited in Figure 4c. Based on the formula, η=a+lgI, where η stands for the overpotential and I represents the current density [35], the measurement results of Tafel curves of the v-FeOOH electrode exhibit a larger exchange current density (3.24 mA cm^−2^) and higher current response than the FeOOH electrode (0.011 mA cm^−2^), indicating the superior conversion kinetics afforded by the v-FeOOH electrode. Besides, the enhanced charge transport of the v-FeOOH electrode is also verified by the EIS spectra before cycling (Figure 4d). All of these results indicate that the v-FeOOH electrode could effectively promote the electrochemical kinetics of the sulfur redox reaction. In addition, Li_2_S nucleation experiments on the v-FeOOH and FeOOH electrodes were performed based on potentiostatic discharge technology (Figure 4e,f). In the two electrodes, v-FeOOH has the earliest and highest peak current, illustrating that v-FeOOH can improve the kinetics of Li_2_S nucleation. Furthermore, v-FeOOH delivers the capacity of Li_2_S precipitation (133.49 mAh g^−1^), which is larger than the FeOOH (106.20 mAh g^−1^) electrode. These results demonstrated that v-FeOOH can boost sulfur catalysis [36]. To better understand the particular redox reaction mechanism of Li–S batteries in the charge–discharge processes, we carried out in situ XRD. The device diagram of the in situ XRD test of a lithium sulfur battery is shown in Appendix A. The potentials and XRD patterns were collected in the discharge process and charge process shown in Figure 4g. In Figure 4g, the sharp crystalline α-S_8_ (JCPDS 008-0247) peaks can clearly be found to disappear in the discharge process and slowly appear with the depth of charge [37]. Moreover, a new peak gradually forms at ∼27°, indicating that sulfur is reduced to the Li_2_S phase (JCPDS 023-0369) [38]. The signal of Li_2_S disappears slowly in the process of charge, which forcefully confirms that v-FeOOH can contribute to the conversion of polysulfide [39]. It should be noted that the peaks at around 24° and 25.65° can both be found during the discharge and charge processes, corresponding to the separator material and rGO-CNTs [40,41].

To verify the structural integrity of catalytic material, Figure 5a shows the SEM of the electrodes after 50 cycles. Compared with the FeOOH electrode (Figure 5a right), the v-FeOOH electrode (Figure 5a left) displays a more uniform surface of the cathode. The HRTEM images of the v-FeOOH electrode after cycling show that the FeVs are still retained (Figure 5b,c). TEM elemental mapping presents the even distribution of Fe, O, and S in the electrode after cycling (Figure 5d). Figure 5e shows the CV curves measured with a scanning rate of 0.2 mV s^−1^ of the electrodes after 50 cycles in a voltage range of 1.7–2.8 V. The sharp responding peak current of the v-FeOOH electrode indicates that v-FeOOH advantageously enhances the conversion kinetics of LiPSs redox [42]. The result is in accordance with the EIS measurements for the v-FeOOH and FeOOH electrodes after cycling (Figure 5f). The v-FeOOH electrode exhibits a smaller semicircle and larger slope, which indicates its small interphase contact resistance and fast ion diffusion within the electrode, respectively.

## 4. Conclusions

In conclusion, FeOOH nanosheets with abundant Fe vacancies as the sulfur cathode were introduced into lithium–sulfur batteries. Li–S batteries catalyzed by v-FeOOH exhibited a high initial specific capacity of 1645.5 mAh g^−1^ at 1 C. Moreover, there was still a low capacity fading rate of only 0.033% per cycle after 300 cycles even at 5 C. The good performance is attributed to the following reasons: (1) The graphene and carbon nanotube network can provide a transformation path for electron and lithium ion conduction. (2) FeVs sites in FeOOH nanosheets serve as a lithium ion anchor and sulfur species accelerator through the strong S_x_Li...FeVs interactions. (3) The structure of FeOOH after cycling remains intact and defective guarantees the active sites, leading to the electrochemical stability and efficiency of lithium–sulfur batteries. Therefore, FeOOH with FeVs loading on the graphene and carbon nanotube network makes for fast transport of lithium ion and inhibits the polysulfides’ shuttle. The work we proposed can not only provide a reference for designing transition metal ion defect materials, but also would hold great application potential in enhancing the performance of Li–S batteries.

## Figures and Tables

**Figure 1 nanomaterials-13-00909-f001:**
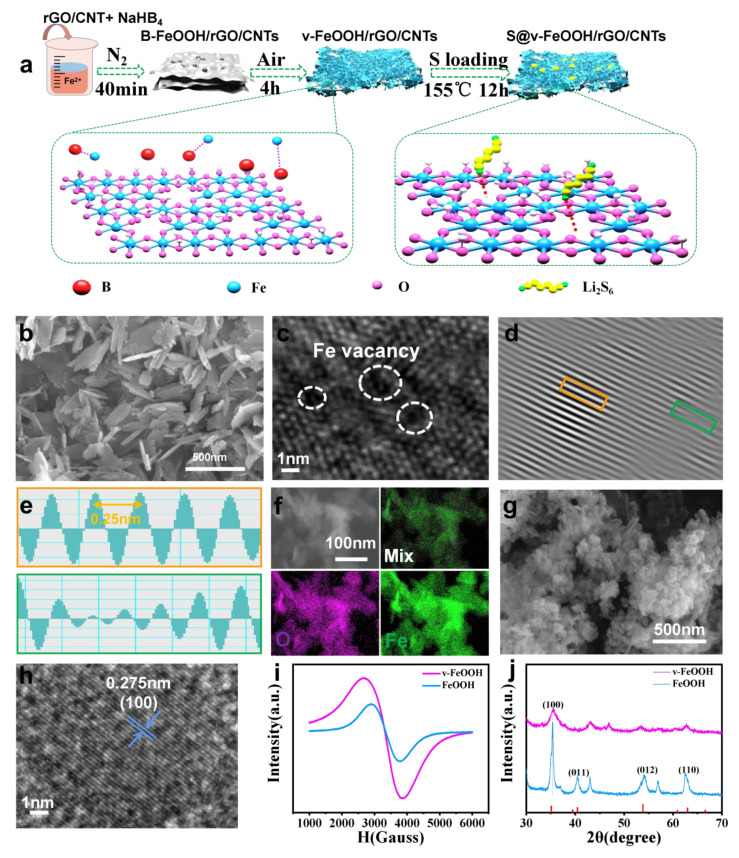
(**a**) The synthesis schematic illustration of the v-FeOOH/rGO/CNTs @S composite. (**b**) SEM and (**c**) HRTEM images of v-FeOOH. (**d**) IFFT of selected area. (**e**) Lattice spacing profiles at the frame of yellow and green. (**f**) High-magnification elemental mapping of v-FeOOH. (**g**) SEM and (**h**) HRTEM images of FeOOH. (**i**) EPR spectra of v-FeOOH and FeOOH. (**j**) XRD patterns of v-FeOOH and FeOOH, respectively.

**Figure 2 nanomaterials-13-00909-f002:**
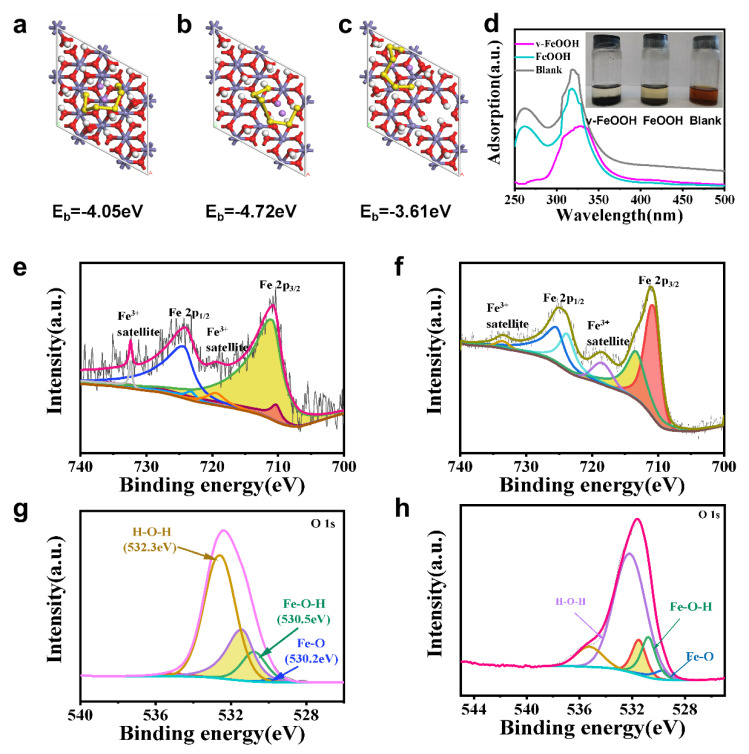
Geometrically stable configurations of L_i2_S_6_ adsorption on (**a**) FeOOH (001), (**b**) v-FeOOH (001), and (**c**) neighboring location to Fe vacancies’ surfaces. The yellow, purple, gray, red, and white balls are S, Li, Fe, O, and H atoms, respectively. (**d**) The digital photos of LiPSs solution adsorption by v-FeOOH and FeOOH after 4 h and the corresponding UV/Vis spectra. (**e**,**f**) Fe 2p XPS spectra of v-FeOOH and FeOOH, respectively. (**g**,**h**) O 1s XPS spectra of v-FeOOH and FeOOH, respectively.

**Figure 3 nanomaterials-13-00909-f003:**
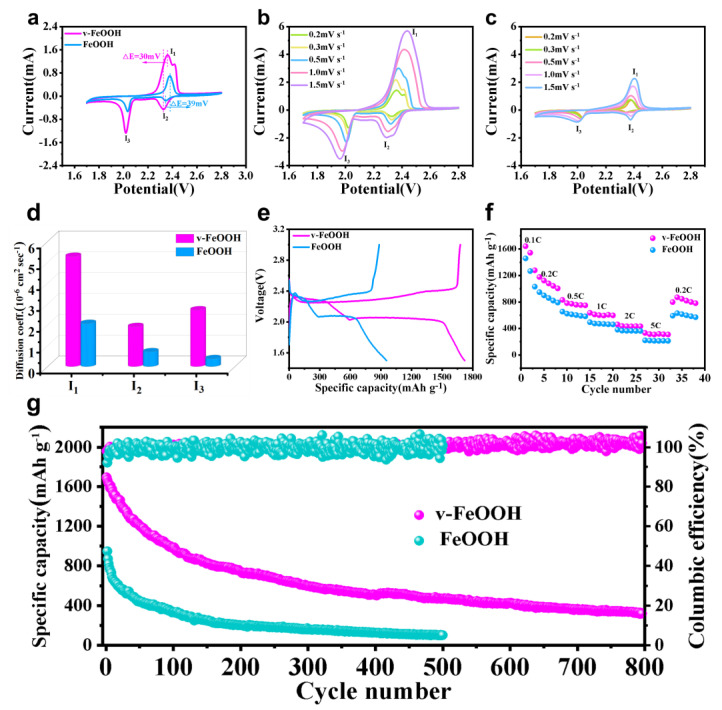
(**a**) Comparison of v-FeOOH and FeOOH electrode cyclic voltammograms tested with a scan rate of 0.2 mV s^−1^. CV test of the (**b**) v-FeOOH and (**c**) FeOOH electrodes at different scan rates. (**d**) Peak current versus square root scan rate for peaks I_1_, I_2_, and I_3_ of the v-FeOOH and FeOOH electrodes. (**e**) Galvanostatic charge–discharge curves at 1 C (of the first cycle). (**f**) Rate properties at different current densities. (**g**) Prolonged cycling stability at 1 C of v-FeOOH and FeOOH electrodes.

**Figure 4 nanomaterials-13-00909-f004:**
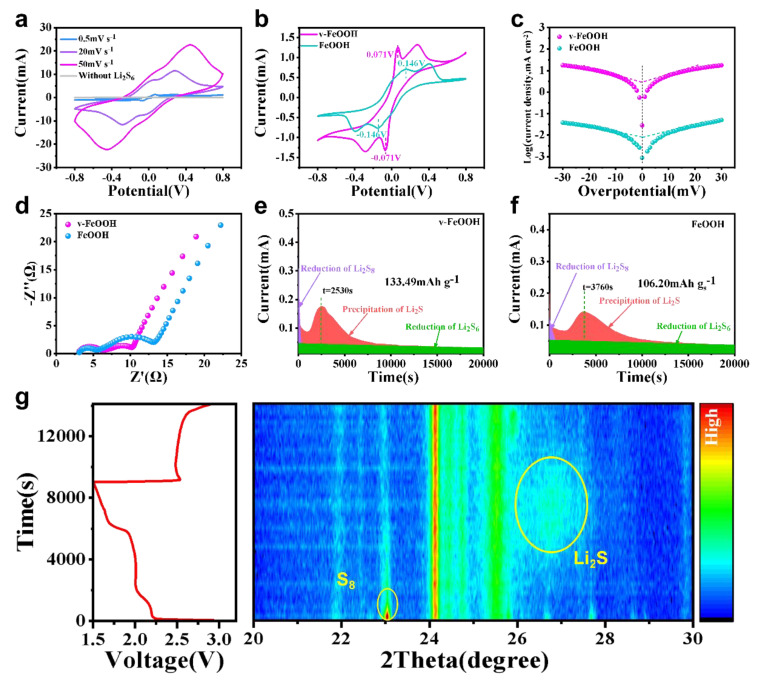
(**a**) CV measurement of symmetric cells for v-FeOOH electrodes at different scan speeds. (**b**) CV tests and (**c**) Tafel plots of symmetric cells for v-FeOOH and FeOOH electrodes at 0.5 mV s^−1^. (**d**) EIS measurements of v-FeOOH and FeOOH electrodes before cycling. (**e**,**f**) Potentiostatic deposition of Li_2_S from Li_2_S_8_ on v-FeOOH and (**f**) FeOOH electrodes. (**g**) Time-resolved XRD patterns at 0.5 C of different discharge states and recharge states for the v-FeOOH electrode.

**Figure 5 nanomaterials-13-00909-f005:**
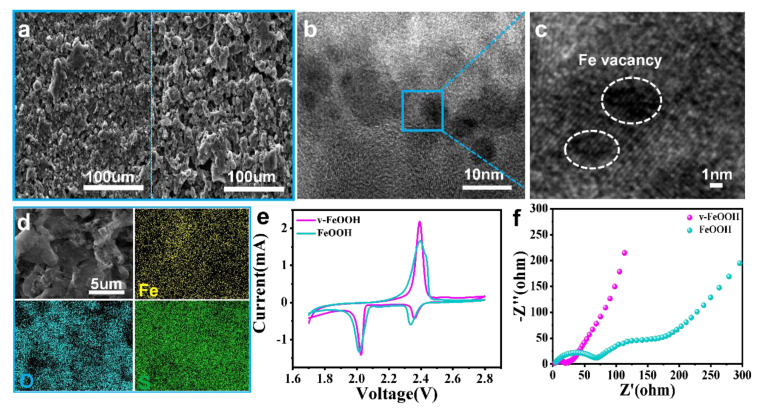
(**a**) SEM images of v-FeOOH and FeOOH after cycling. (**b**,**c**) HRTEM images of the v-FeOOH electrode after cycling. (**d**) High-magnification elemental mapping of the v-FeOOH electrode after cycling. (**e**) CV plots and **(f)** EIS measurements for the v-FeOOH and FeOOH electrodes after cycling.

## Data Availability

Data presented in this article are available upon request from the corresponding author.

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
