# Peer review of "Cation Vacancies in Feroxyhyte Nanosheets toward Fast Kinetics in Lithium–Sulfur Batteries"

_nanomaterials, 2023, doi:10.3390/nano13050909_

Round 1
Reviewer 1 Report
The manuscript titled “Cation Vacancies in Feroxyhyte Nanosheets toward Fast Kinetics in Lithium-Sulfur Batteries” can be accepted for publication after the following minor revisions.
1. There are several typos and grammatical mistakes in the manuscript. Perform a detailed editing.
2. Other types of rechargeable batteries and importance of energy storage needs to be emphasized in the introduction before discussing Li-S batteries.
3. Perform the first cycle coulombic efficiency of Li-S cells containing cation vacant and control sample
Reviewer 2 Report
The manuscript reports the synthesis of FeOOH / rGO/CNTs @ S composite active material, in which the FeOOH presents Fe vacancies, which are active catalysts for lithium polysufides electrochemical reaction in the Lithium- Sulfur battery. Although similar materials have already been used in Li-S batteries [Preparation and Application of Nanorod FeOOH/CNT@S Composites for High-Performance Lithium−Sulfur Batteries, ACS Appl. Energy Mater. 2021, 4, 8368−8376]; the idea of introducing Fe vacancies is well explained. The cathode shows faster kinetics than that based on FeOOH / rGO/CNTs @ S composite with no Fe vacancies and reduced shuttle effect. The structural and morphological characterization of the material is detailed. The catalytic activity of the composite material is demonstrated by cyclic voltammetry, UV-spectroscopy, Tafel plots and potentiostatic deposition of Li2S and finally supported by DFT calculations.
The evaluation of the cathode performance in the Li-S cell are unconvincing and in some way contradictory.
Fig3c shows the rate capability of the cell containing the two different cathodes. At the 15th cycle, the specific capacity at 1C is less than 400 mAh/g for the cell containing the v FeOOH / rGO/CNTs @ S. At the same C rate (1C) the specific capacity of the cell is more than 1000 mAh/g at the 100th cycle (Figure 3g). I can understand that the test conditions are different. In the long cycling test at 1C the shuttle effect is less since the early cycles, due to the high C rate. However, I still don't understand such a big difference in the capacity values of v FeOOH / rGO/CNTs @ S at the same C rate in Fig 3c and Fig 3g . Can the Authors explain this?
At which cycle the voltage profiles in Figure 3e were obtained?
Moreover, for both v FeOOH and FeOOH at 0.1C and 0.2C rates, ( Figures S12 and S13), the charge curves present an anomalous profile, with a plateau at potentials greater than 2.4V. Can the Authors explain this behavior? In Figure S15 the coulombic efficiency is always higher than 100%. Why?
Line 200 page 6: “…and capacity retention of 320.8 mAh g-1 “ : This is not a “capacity retention”
Reviewer 3 Report
Li-S batteries were studied by manufacturing FeOOH nanosheets with rich Fe vacancies. The research idea conformed to the scientific theory, the experiments were conducted very systematically, and the research results were very excellent, so it is highly recommended for publication in this journal.
Author Response
Thanks for your review.